# Creating Robust Deep Neural Networks With Coded Distributed Computing for IoT Systems

## ABSTRACT

The increasing interest in serverless computation and ubiquitous wireless networks has led to numerous connected devices in our surroundings. Among such devices, IoT devices have access to an abundance of raw data, but their inadequate resources in computing limit their capabilities. Specifically, with the emergence of deep neural networks (DNNs), not only is the demand for the computing power of IoT devices increasing but also privacy concerns are pushing computations towards the edge. To overcome inadequate resources, several studies have proposed the distribution of work among IoT devices. These promising methods harvest the aggregated computing power of the idle IoT devices in an environment. However, since such a distributed system strongly relies on each device, unstable latencies, and intermittent failures, the common characteristics of IoT devices and wireless networks, cause high recovery overheads. To reduce this overhead, we propose a novel robustness method with a close-to-zero recovery latency for DNN computations. Our solution never loses a request or spends time recovering from a failure. To do so, first, we analyze the underlying matrix-matrix computations affected by distribution. Then, we introduce a new coded distributed computing (CDC) method that has a constant cost with the increasing number of devices, unlike the linear cost of modular redundancies. Moreover, our method is applied in the library level, without requiring extensive changes to the program, while still ensuring a balanced work assignment during distribution. To illustrate our method, we perform experiments with distributed systems comprising up to 12 Raspberry Pis.

## 1 INTRODUCTION

Recent years have witnessed an emergence of deep neural networks (DNNs) applications and their represented services [1]. Additionally, with the proliferation of Internet-of-Things (IoT) devices, these devices are inseparable from our daily lives [2–4]. The conventional methods to process raw IoT data are to offloaded them to cloud services [5, 6]. However, moving such a tremendous amount of data incurs a high amount of monetary cost and delay, besides creating a major concern of privacy leakages. Therefore, to provide a solution to this challenge and meet the demand for data explosion, serverless and edge computation paradigms are recognized as a promising solution [7–9]. As a result, pushing the frontier of DNNs computations to the edge is receiving a tremendous amount of interest both from academia with new exciting methods [7, 9–15] and from the industry with commercial edge-tailored hardware accelerators [16–18].

Processing the IoT data locally in-the-edge and on each individual device may suffer from poor performance and energy efficiency [7, 9]. This is because the demanded computational power

*Conference'17, July 2017, Washington, DC, USA*
2020. ACM ISBN 978-x-xxxx-xxxx-x/YY/MM. . . $15.00
https://doi.org/00.0000/0000000.0000000

from resource-hungry DNN-based applications outweighs the computation capacity and energy constraints of IoT devices. Furthermore, the computational demands are escalated because these devices have to meet real-time and time-sensitive constraints while processing raw information that is directly gathered from their environment. Even for edge-tailored hardware accelerators, it has been shown that the real timeliness of applications is not guaranteed because of a wide variety in machine learning frameworks and DNN applications [19, 20]. Nevertheless, privacy concerns [21, 22], unreliable connection to the cloud, tight real-time requirements, and personalization are still pushing inferencing to the edge.

To address the mentioned resource constraint challenges, a promising solution is the distribution of heavy computations among idle devices present in the environment [10–12, 15]. This is because the state-of-the-art IoT networks are formed with various IoT sensors and recording agents, such as HD cameras and temperature sensors, many of which are capable of performing computations. Moreover, for any given time, some IoT devices are idle. However, such a distribution is susceptible to failures, from short disconnectivity and user interaction to losing a device. With these intermittent or permanent failures, we may lose valuable time-sensitive and real-time information. This fact necessitates developing a robust method for tolerating these failures. Additionally, since IoT networks use wireless technology, unreliability and variability in their networks are much higher than acceptable limits to ensure a robust system.

This work extends current studies that enable distributed single-batch inference of DNNs in the edge [10–12, 15] to tolerate failures with close-to-zero recovery latency. To do so, first, we analyze general methods of distributing the computations of DNNs (with a focus on convolution neural networks (CNNs)) and how their underlying matrix-matrix computations are affected by distribution. Such a detailed study is necessary to introduce a general seamless method within the underlying library or machine learning framework. Then, we propose a new recovery method based on coded distributed computing (CDC) that enables distributed DNN models on IoT devices to tolerate failures and not lose time-sensitive and real-time information. Our method is inspired by CDC applications in big data analytics [23, 24], and speeding up distributed learning using codes [25]. In summary, these works *theoretically* analyze CDC methods that reduce latency by increasing computation (See related work, Section 8).

To introduce robustness in distributed IoT systems, we introduce an extra coded computation per device. The introduced extra computations are derived by thoroughly studying various distribution techniques in their underlying matrix-matrix computations for inference operation in DNNs. These extra computations are similar in nature to those of DNNs, which ease balancing the work among IoT devices and reduce the deployment cost. Such a balanced distribution is essential in attaining the expected performance. Additionally, since our method is implemented in the underlying matrix-matrix

computations of DNN layers, it does not require extensive changes to the user's program and is implemented at the library level. Moreover, our method, even at the time of failures, provides close-to-zero recovery time, which is necessary for critical time-sensitive tasks. This is in contrast with approaches that sacrifice latency for robustness to recompute the missing part of the data. Finally, compared with conventional modular redundancy methods, that introduce redundancy in computation by introducing a linear number of additional devices, our method has a constant cost with the increasing number of devices. We demonstrate our method on distributed systems comprising of Raspberry Pis (RPis), which represent the de facto choice for several small and edge use cases.

In summary, this is the first work, to the best of our knowledge, that improves robustness in distributed IoT systems, with the following contributions:

- We thoroughly analyze how general methods of distributing the computation of DNNs affect the underlying matrix-matrix computations.
- We propose a novel fault recovery method based on CDC that has close-to-zero recovery latency, does not disturb the balanced work assignment in distribution, requires minimal changes to the user's program, and has a constant cost with the increasing number of devices.
- We demonstrate our method on distributed systems of up to 12 Raspberry Pis and report our experimental results.

## 2 MOTIVATION

Distributed IoT systems are a good candidate for creating in-the-edge computing domains for the user-space applications. This approach has many advantages; for instance, since the collected raw data remains local, user privacy is less exposed and vulnerable to attacks. Furthermore, such a platform does not depend on the network availability and does not require expensive quality-of-service guarantees [26, 27] for several time-sensitive applications. Additionally, several applications in robotics [28–30] and unmanned aerial vehicles (UAVs) [31, 32] benefit from such systems. Nevertheless, the disadvantage is that a connected IoT system is a dynamic system and some devices may unexpectedly become busy or lose their connection. In a distributed design, in which each device is important, such failures are destructive. For instance, the system may go offline for a long duration or accuracy may drop significantly. Thus, the robustness of the entire system becomes a concern, specifically when users may rely on this system for many sensitive and time-critical applications. Moreover, the robustness issue is exacerbated by the local wireless networks of these systems, causing the latency between the devices to be unreliable and unstable.

To illustrate unreliability in the communication latency of IoT systems, Figure 1 shows a histogram of the arrival times for data

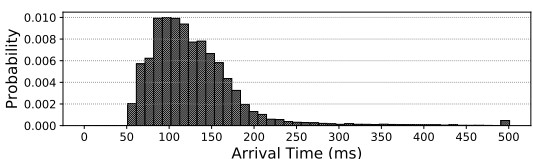

**Figure 1: Arrival time histogram of data packets in a WiFi network for a four-device IoT system with RPis.**

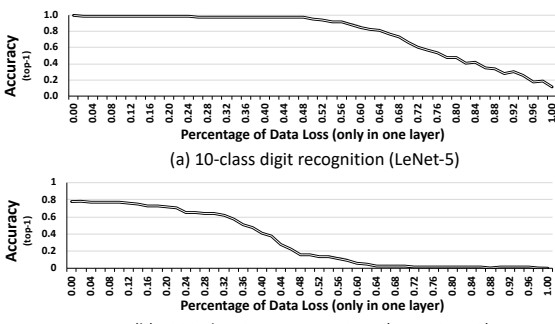

(a) 10-class digit recognition (LeNet-5)

(b) 1000-class image recognition (Inception v3)

**Figure 2: High percentage data loss, common in distributed IoT systems, causes destructive accuracy drops.**

packets in a four-device IoT system with four RPis [33] (see system setup in Section 6). This system performs the computation for a fully-connected layer of size 2048 in a distributed fashion and waits for the response. The measured time for the computation of a fully-connected layer of size 2048 on a single device is 50 ms. This is why, in Figure 1, no packet arrives earlier than 50 ms. As seen, around 34% of the arrival times is within 100 ms, and 42% is within 150 ms. So, even after 2x the computation time, around 34% of the packets have not arrived yet. Such behavior in distributed systems causes *straggler problem*, in which the slowest node in the distributed system defines the total latency. Our method, by introducing robustness in such systems, can additionally alleviate the straggler problem while also guaranteeing close-to-zero recovery latency.

To understand how failures are destructive in DNN applications, we perform another set of experiments, in which some part of data within a layer is lost. We choose two models: LeNet-5 [34] and Inception v3 [35]. LeNet-5 is a simple model that detects handwritten digits from 10 classes and consists of only five layers. On the other hand, Inception v3 is a modern DNN model for image recognition for 1k classes with 159 layers. Figure 2 illustrates the accuracy drop in these models when some part of the data in a layer is lost. As seen, for large percentages of data loss (> 70%) per layer that are common in distributed IoT systems, the accuracy drop is destructive. Additionally, by comparing Figures 2a and b, we see that that the sensitivity to data loss in more generalized models will only become worse. Since the amount of data loss happens in larger granularities, the current robustness methods in DNNs (e.g., bit-level tolerance [36]) are insufficient to recover the loss. In contrast, our proposed robustness method is designed specifically for such a high amount of data loss and can recover from it with close-to-zero latency.

## 3 DNN COMPUTATIONS

Since for implementing the CDC-based robustness, we need to know how the underlying matrix-matrix multiplications are derived, this section provides a summary overview of computations within DNNs. Area experts may skip forward to the next section. In details, first, we discuss the fully-connected layer that is prevalent in types of DNNs (MLP, RNN, LSTM, and CNNs). Second, we overview the convolution layer, used in CNNs. These layers are the most compute- and data-intensive layers in the mentioned

DNNs [37]. We also describe how these computations are done in underlying matrix-matrix multiplication libraries (GEMM).

**Fully-Connected Layer:** A fully-connected layer has several outputs, each of which could be written as

$$a_j^l = \sigma\Big(\sum_k w_{jk}^l a_k^{l-1} + b_j^l\Big), \tag{1}$$

in which $a_j^l$ is the $j^{\text{th}}$ activation or input of the $l^{\text{th}}$ layer, $w_{jk}^l$ is the weight from $k^{\text{th}}$ input in the $(l-1)^{\text{th}}$ layer to the $j^{\text{th}}$ output in the $l^{\text{th}}$ layer, $b_j^l$ is the bias of the $j^{\text{th}}$ output in the $l^{\text{th}}$ layer, and $\sigma$ is the activation function such as ReLU ($max(0, x)$). Since the notation of $w_{jk}^l$ is from $k$ to $j$, we can write the computations of the $l^{\text{th}}$ layer as the below matrix operation:

$$\begin{bmatrix} w_{11} & w_{12} & \cdots & w_{1k} \\ w_{21} & w_{22} & \cdots & w_{2k} \\ \vdots & \vdots & \ddots & \vdots \\ w_{m1} & w_{m2} & \cdots & w_{mk} \end{bmatrix}_{m \times k} \times \begin{bmatrix} a_1' \\ a_2' \\ \vdots \\ a_k' \end{bmatrix}_{k \times 1} = \begin{bmatrix} a_1 \\ a_2 \\ \vdots \\ a_m \end{bmatrix}_{m \times 1}, \tag{2}$$

in which $m$ is the number of output elements, $k$ is the number of input elements, and $a'$ represents previous layer activations. Or, we can write the computations in the $l^{\text{th}}$ layer as

$$\mathbf{a}^l = \sigma(\mathbf{W}^l \mathbf{a}^{l-1} + \mathbf{b}^l). \tag{3}$$

In training, the learning is done by adjusting $\mathbf{W}$ and $\mathbf{b}$. Furthermore, note that the computations of Equation 3, in its current format, can easily utilize GEMM libraries [38, 39]. Thus, no transformation is needed to execute fully-connected layers.

**Convolution Layer:** CNN models predominantly process visual data with convolution layers (conv). A convolution layer applies the *same* set of weights or filters similar to fc, but to subsets or patches of input. Figure 3 depicts the convolution of an input with size $H_i \times W_i \times C_i$ with $K$ square filters of size $F \times F \times C_i$. Each filter creates a channel of the output. Thus, the depth of the output, $C_o$, is equal to the number of filters, $K$, $C_o = K$. The height and width of the output are determined by how a filter is swept across the input by parameters such as stride ($s$), filter size ($f$), and padding ($p$). In short, the output size in any dimension is derived from $\lfloor \frac{i-f+2p}{s} \rfloor$, in which $i$ is the input size in that dimension. For the sake of simplicity, in this paper, we assume the same padding condition.

To perform *conv*, almost all machine learning frameworks perform a transformation to harness the power of extensively optimized parallel GEMM libraries [38, 39]. Our robustness method is also implemented in this level to minimize changes to the user's program. The essence of the transformation is to unroll the input patches (a 3D matrix) and filters (a 4D matrix) in 2D in a way that a single matrix-matrix multiplication produces the unrolled version of the output in 2D. To do so, the weight matrix of the filters,

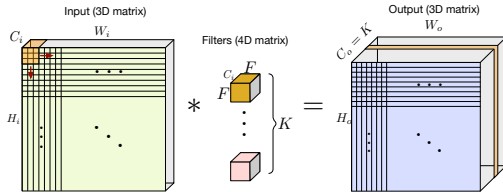

**Figure 3: Visualization of a convolution layer; each filter creates a depth channel in the output.**

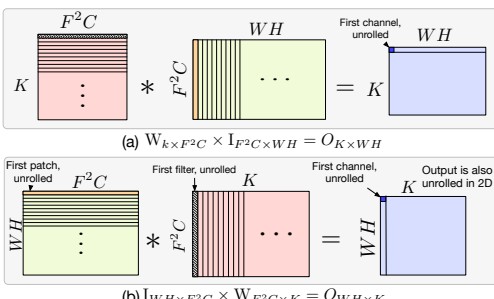

(a) $\mathrm{W}_{k \times F^2 C} \times \mathrm{I}_{F^2 C \times WH} = \mathrm{O}_{K \times WH}$

(b) $\mathrm{I}_{WH \times F^2 C} \times \mathrm{W}_{F^2 C \times K} = \mathrm{O}_{WH \times K}$

**Figure 4: Underlying transformation of a convolution operation to use GEMM libraries [38, 39].**

$F \times F \times C \times K$, is converted to a $K \times F^2 C$ matrix. Similarly, the input matrix of size $W \times H \times C$ is converted to a $F^2 C \times WH$ matrix by unrolling each patch and repeating the overlapping elements (if necessary). Figure 4a illustrates the unrolling operation we discussed, besides another possible way in Figure 4b, which transforms convolution as matrix-matrix multiplications as below:

$$\mathbf{O}_{K \times WH} = \mathbf{W}_{K \times F^2 C} \times \mathbf{I}_{F^2 C \times WH}. \tag{4}$$

**Other Layers:** Other layers such as max-pooling (maxpool) or average-pooling (avgpool) layers have relatively lower computation demand compared to fully-connected and convolution layers. Hence, we group them with their parent layers.

## 4 DNN DISTRIBUTION IN IOT SYSTEMS

A variant of model-parallelism methods [40, 41], specifically targeting single-batch inferences can be exploited [12] to distribute the DNN computations. This is because the new constraints introduced by the edge devices change the assumptions of the current cloud-based systems. In short, these new constraints are (i) a limited number of local requests and real-time requirements, which enforces single-batch inferences instead of batching for performing data-level parallelism, and (ii) the limited compute power of IoT devices coupled with real-time system constraints, which renders workstation-based parallelism/distribution methods for processing DNNs less efficient in IoT devices. Thus, distributions based on model-parallelism are ideal for IoT devices that suffer from limited resources but need to satisfy real-time constraints. In the following, we introduce distribution methods for fully-connected and convolution layers.

**Distribution for Fully-Connected Layers:**

- ***Output Splitting:*** In this method, creating output is divided among devices, shown in Figure 5a. For each activation, its whole computation is performed on one device. To do this, we need all the input elements. Finally, when every device is done, a merge operation concatenates the output elements.
- ***Input Splitting:*** In this method, each device computes a part of the entire output, shown in Figure 5b. To do so, a part of the input is transmitted and each device computes all parts of the output that are dependent on the received input. Finally, when every device is done, a merge operation executes the summation part.

**Distribution Methods for Convolution Layers:**

- ***Channel Splitting:*** In this method, each device only handles a set of filters and creates a part of the depth dimension

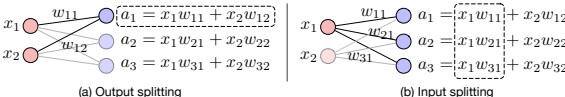

Figure 5: Fully-connected layer model parallelism methods.

in the output. Thus, each device only needs the weights of its dedicated filters but requires all the input data. The final merging operation consists of simply concatenating the individual outputs (either before or after activation function).

- **Spatial Splitting:** In this method, the input is divided spatially, including the overlap element between input patches, and each device processes only some part of the input for all the filters. Thus, each device needs all the filter weights. Finally, the merge operation is a concatenation on both height and width dimensions.

- **Filter Splitting:** In this method, both the filters and input are divided among devices in the depth dimension. Each device only processes the corresponding sets of input and filters. Thus, each device computes a partial sum of all output elements. Finally, the merging operation consists of a summation and application of the activation function.

## 5 ROBUSTNESS WITH CDC

This section first describes how distribution methods change the computation of each device from the view of underlying matrix-matrix computation. Such analysis helps us to easily generalize our method and apply it at the library level. Next, we provide a simple example of our robustness method that handles only one output per device. Then, we generalize our method to multiple outputs per device. Finally, we study the suitability of the distribution methods.

### 5.1 Distribution and Matrix Operations

**Fully-Connected Layer:** A fully-connected layer performs Equation 3 with GEMM. First we consider the matrix-matrix multiplication part, or $\mathbf{W}^l \mathbf{a}^{l-1}$. Figure 6 illustrates how output splitting affects weight and output matrices for an example with four devices. Since each device calculates a set of separate outputs, the output matrix is created separately by each device (and concatenated later). Such separation in output generation also divides the weight matrix along the y-axis, which has a one-by-one relationship with the output matrix division. Each device needs a copy of the input matrix, and the input matrix is not divided.

In the input-splitting method, as Figure 7 depicts for the same four-devices example, the input matrix is divided between the devices. Similarly, the weights corresponding to those inputs are also divided along the x-axis among devices. Each device calculates partial sums for the entire output elements. Finally, all partial sums are aggregated to create the final output. Regarding bias and the activation function, we can extend the above reasoning. For output

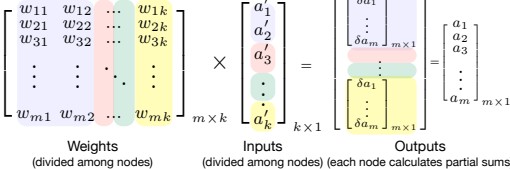

Figure 7: Distribution of input splitting for fully-connected layers.

splitting, biases and the activation function can also be divided among the devices. But, for input splitting, both need to be applied after the aggregation. Since the majority of the computation time of DNNs is spent on matrix-matrix multiplications, such a difference does not have a big impact on computation time.

**Convolution Layer:** By utilizing Figure 4 for convolution layer, the channel-splitting method divides the filter weight matrix along the y-axis, as Figure 8 shows for two devices. Likewise, since the output is unrolled, such division translates to a similar along-the-y-axis division of the output matrix. Hence, channel splitting in convolution layers is the same as output splitting into fully-connected layers and any robustness analysis is applicable on both, but with a different set of weights and inputs (i.e., unrolled version of filters and patches in convolution layers).

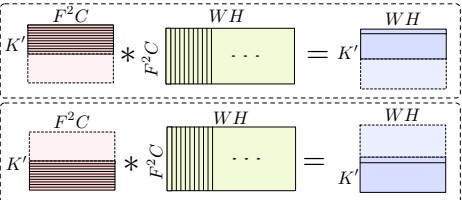

Figure 8: Distribution of channel splitting for convolution layers.

In the spatial-splitting method, since each input patch is unrolled column-wise in the input matrix when we spatially divide the input, this division translates to an along-the-x-axis division of the input matrix. However, unlike input splitting in fully-connected layers, filter weights cannot be divided. Therefore, spatial splitting, as conceptually shown in Figure 9 for two devices, divides the input matrix of Equation 4 along the x-axis.

In the filter-splitting method, a close representation of input splitting for fully-connected layers, both filter weights and input are divided depth-wise. Since both filter weights and input are unrolled, we need to divide the weight and input matrices along the x- and y-axes, respectively. This distribution is similar to the outer product approach in matrix multiplication, versus the most commonly known algorithm of the inner product approach. Figure 10 shows this approach with two devices. Each device produces a partial sum for the entire elements. To create the final output, the final device needs to aggregate all the elements and apply the activation function.

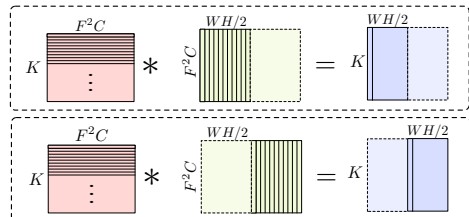

Figure 9: Distribution of spatial splitting for convolution layers.

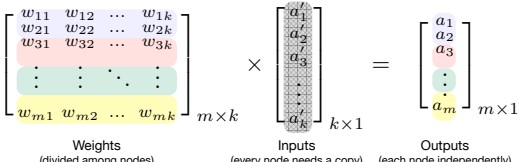

Figure 6: Distribution of output splitting for fully-connected layers.

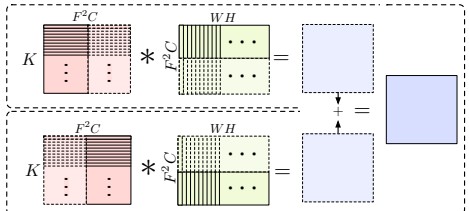

**Figure 10: Distribution of filter splitting for convolution layers.**

## 5.2 Robustness: A Simple Example

We present a simple example of our CDC-based robustness to facilitate understanding. Consider a fully-connected layer with two input and output elements, written as:

$$\begin{bmatrix} w_{11} & w_{12} \\ w_{21} & w_{22} \end{bmatrix} \times \begin{bmatrix} a'_1 \\ a'_2 \end{bmatrix} = \begin{bmatrix} a_1 \\ a_2 \end{bmatrix}. \tag{5}$$

Assume that we perform output splitting. Now, by adding a row to the weight matrix with the value of $[w_{11} + w_{21} \; w_{12} + w_{22}]$, we can create the summation of two outputs, or $a_1 + a_2$. Therefore, with such addition, the above equation becomes:

$$\begin{bmatrix} w_{11} & w_{12} \\ w_{21} & w_{22} \\ w_{11} + w_{21} & w_{12} + w_{22} \end{bmatrix} \times \begin{bmatrix} a'_1 \\ a'_2 \end{bmatrix} = \begin{bmatrix} a_1 \\ a_2 \\ a_1 + a_2 \end{bmatrix}. \tag{6}$$

Since the summation of the weights can be done offline and is not dependent on inputs, we can rewrite about equation as:

$$\begin{bmatrix} w_{11} & w_{12} \\ w_{21} & w_{22} \\ w^{cdc}_{:1} & w^{cdc}_{:2} \end{bmatrix} \times \begin{bmatrix} a'_1 \\ a'_2 \end{bmatrix} = \begin{bmatrix} a_1 \\ a_2 \\ a^{cdc} \end{bmatrix}. \tag{7}$$

The newly added weights to the weight matrix are the column sums of the weight matrix that is done offline before loading the weights. Now, with the addition of another device, we can guarantee to recover from one missing output with only a local subtraction in the final device. This method has three main benefits:

- First, this level of guarantee on all devices is just with an addition of one device, compared to a double modular redundancy method that duplicates all devices.
- Second, this method is faster than redoing all operations since the subtraction of two local values that we already have received is almost immediate than restarting all operations. This is because, the vanilla recovery method consists of loading a set of new weights (corresponding to the missing values) in the final device, asking for the input from previous devices, and performing multiplications with all of its associated overhead of communication.
- Third, although we introduced the computations corresponding to $a^{cdc}$, these computations are similar in nature to the computations of $a_1$ and $a_2$. Hence, the distribution of these newly added computations follows the same rules and would not create an imbalance in the modified distribution.

## 5.3 Generalization of Robustness

In this section, we extend our simple scenario, in which each device computes only one output element, to a more realistic scenario, in which each device computes hundreds of elements. Similarly, we showcase the output-splitting method as our example. Assume a fully-connected layer performing the below equation:

$$\begin{bmatrix} w_{11} & w_{12} & \dots & w_{1k} \\ w_{21} & w_{22} & \dots & w_{2k} \\ \vdots & \vdots & \ddots & \vdots \\ w_{m1} & w_{m2} & \dots & w_{mk} \end{bmatrix}_{m \times k} \times \begin{bmatrix} a'_1 \\ a'_2 \\ \vdots \\ a'_k \end{bmatrix}_{k \times 1} = \begin{bmatrix} a_1 \\ a_2 \\ \vdots \\ a_m \end{bmatrix}_{m \times 1}. \tag{8}$$

By distributing the computations among two devices, each of the devices perform the computations for $m/2$ of output elements. The computations per each device are

$$\begin{bmatrix} w_{11} & w_{12} & \dots & w_{1k} \\ w_{21} & w_{22} & \dots & w_{2k} \\ \vdots & \vdots & \vdots & \ddots & \vdots \\ w_{\frac{m}{2}1} & w_{\frac{m}{2}3} & \dots & w_{\frac{m}{2}k} \end{bmatrix}_{\frac{m}{2} \times k} \times \begin{bmatrix} a'_1 \\ a'_2 \\ \vdots \\ a'_k \end{bmatrix} = \begin{bmatrix} a_1 \\ a_2 \\ \vdots \\ a_{\frac{m}{2}} \end{bmatrix}, \text{and} \tag{9}$$

$$\begin{bmatrix} w_{(\frac{m}{2}+1)1} & w_{(\frac{m}{2}+1)2} & \dots & w_{(\frac{m}{2}+1)k} \\ w_{(\frac{m}{2}+2)1} & w_{(\frac{m}{2}+2)2} & \dots & w_{(\frac{m}{2}+2)k} \\ \vdots & \vdots & \vdots & \vdots \\ w_{m1} & w_{m2} & \dots & w_{mk} \end{bmatrix} \times \begin{bmatrix} a'_1 \\ a'_2 \\ \vdots \\ a'_k \end{bmatrix} = \begin{bmatrix} a_{(\frac{m}{2}+1)} \\ a_{(\frac{m}{2}+2)} \\ \vdots \\ a_m \end{bmatrix} \tag{10}$$

in which input matrices are the same, but the weight matrix is divided along the y-axis. Each device creates separate parts of the output matrix. To introduce robustness, the new weight matrix would be as follows:

$$\begin{bmatrix} w_{11}+w_{(\frac{m}{2}+1)1} & w_{12}+w_{(\frac{m}{2}+1)2} & \dots & w_{1k}+w_{(\frac{m}{2}+1)k} \\ w_{21}+w_{(\frac{m}{2}+2)1} & w_{22}+w_{(\frac{m}{2}+2)2} & \dots & w_{2k}+w_{(\frac{m}{2}+2)k} \\ \vdots & \vdots & \ddots & \vdots \\ w_{\frac{m}{2}1}+w_{m1} & w_{\frac{m}{2}2}+w_{m2} & \dots & w_{\frac{m}{2}k}+w_{mk} \end{bmatrix}_{\frac{m}{2} \times k}. \tag{11}$$

By multiplying this new weight matrix with inputs, the below output matrix is created:

$$\begin{bmatrix} a_1 + a_{(\frac{m}{2}+1)} \\ a_2 + a_{(\frac{m}{2}+2)} \\ \vdots \\ a_{\frac{m}{2}} + a_m \end{bmatrix}_{\frac{m}{2} \times 1}, \tag{12}$$

which is is the summation of two output matrices in Equations 9 and 10. Therefore, by introducing such a weight matrix as Equation 11, we can introduce robustness. Similar to our simple example, the computation of this new weight is done offline, recovery has a close-to-zero latency, the robustness covers all devices, and the new computation does create an imbalanced distribution.

In contrast, splitting methods that work based on dividing the input matrix among the devices does not yield similar benefits. To illustrate why, we study input splitting among two devices for the computation of the same fully-connected layer presented in Equation 8. Input splitting for fully-connected layers divides the input and the weight matrix along the x-axis. Accordingly, the computations per device are:

$$\begin{bmatrix} w_{11} & w_{12} & \dots & w_{1\frac{k}{2}} \\ w_{21} & w_{22} & \dots & w_{2\frac{k}{2}} \\ \vdots & \vdots & \ddots & \vdots \\ w_{m1} & w_{m2} & \dots & w_{m\frac{k}{2}} \end{bmatrix}_{m \times \frac{k}{2}} \times \begin{bmatrix} a'_1 \\ a'_2 \\ \vdots \\ a'_{\frac{k}{2}} \end{bmatrix}_{\frac{k}{2} \times 1} = \begin{bmatrix} \delta a_1 \\ \delta a_2 \\ \vdots \\ \delta a_m \end{bmatrix}_{m \times 1} \tag{13}$$

$$\begin{bmatrix} w_{1(\frac{k}{2}+1)} & w_{1(\frac{k}{2}+2)} & \dots & w_{1k} \\ w_{2(\frac{k}{2}+1)} & w_{2(\frac{k}{2}+2)} & \dots & w_{2k} \\ \vdots & \vdots & \ddots & \vdots \\ w_{m(\frac{k}{2}+1)} & w_{m(\frac{k}{2}+2)} & \dots & w_{mk} \end{bmatrix}_{m \times \frac{k}{2}} \times \begin{bmatrix} a'_{\frac{k}{2}+1} \\ a'_{\frac{k}{2}+2} \\ \vdots \\ a'_k \end{bmatrix}_{\frac{k}{2} \times 1} = \begin{bmatrix} \delta a_1 \\ \delta a_2 \\ \vdots \\ \delta a_m \end{bmatrix}_{m \times 1}. \tag{14}$$

**Table 1: Distribution Techniques Suitable for Robustness.**

| Layer | Model-Parallelism Method | Divides Input | Divides Weight | Divides Output | Suitable for Robustness |
|-------|-------------------------|---------------|----------------|----------------|-------------------------|
| fc | Output | ✗ | ✓ | ✓ | Yes |
|    | Input | ✓ | ✓ | ✗ | No |
| conv | Channel | ✗ | ✓ | ✓ | Yes |
|      | Spatial | ✓ | ✗ | ✓ | No |
|      | Filter | ✓ | ✓ | ✓ | No |

Each of the above equations calculates a partial sum. However, as seen, no share factor exists between the two computations. Therefore, if a third device wants to perform coded distribution, it needs to perform the entire calculations of Equations 13 and 14. Such an approach creates unbalanced work between the devices, and has no advantage over just replicating the entire work as modular redundancy methods do.

**Distribution Techniques Suitable for Robustness:** For the distribution methods we introduced in Section 4, based on the aforementioned discussion, only some methods are suitable for our CDC-based robustness. Such suitable methods do not split the input elements but split the weights. Table 1 provides a summary of all the presented methods and whether they are suitable for robustness. For fully-connected layers, the output-splitting method is suitable for robustness. For convolution layers, the channel-splitting method has similar characteristics. Unfortunately, the rest of the distribution methods are not suitable for robustness. This is because to introduce robustness, these methods need to actually perform the entire computation again, which including the communication overhead. For instance, in spatial splitting, although every device has all the weights, they only own some part of the input. Therefore, with our technique, we need another device performing the computation based on the summation of the input parts. Since input elements change, computing such a summation has an overhead during the runtime (2x compute). The filter-splitting method also suffers from the fact that no element from the input or weights is shared between computing devices.

## 6 EXPERIMENTS

This section first presents our system setup and necessary information. Then, we provide two case studies for fault recovery. Our experiments show how our method helps systems to achieve higher performance because of the straggler problem mitigation. Finally, we also compare the coverage to failures of the whole systems with CDC and double modular redundancy.

**Experiments Setup:** We evaluate our method on a distributed system with RPi [33] with 1.2 GHz Quad Core ARM Cortex-A53 CPU and a 900 MHz 1 GB RAM LPDDR2 memory. We choose RPi because they represent the de facto choice for several IoT and edge use cases, they are readily available, and they allow common software packages. Our implementation is created with a software stack based on Docker containers. We use Keras 2.1 [42] with the TensorFlow backend (version 1.5) [43]. For RPC calls and serialization, we use Apache Avro [44]. A local WiFi network with the measured bandwidth of 94.1 Mbps and a measured client-to-client latency of 0.3 ms for 64 B is used.

**Task Creation & Assignment:** The policy of task creation in IoT-based distributed DNN systems is done with either profiling or heuristics that use common monitoring/managing tools such as

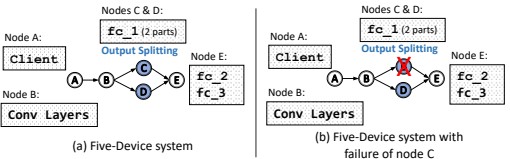

**Figure 11: Case study I: AlexNet on a five-device system.**

Kubernetes. The policies create tasks per device for a given DNN architecture by studying its memory footprint, computation requirement, and communication overhead. Regardless of the policy that finds the optimal distribution (out of scope of this paper, see [12]), all the pre-trained weights are loaded to each device storage so that a device can switch its assigned task easily if needed. For each number of available devices, a single task allocation file is loaded to all devices and each device performs its allocated tasks based on the file. In our implementation, we use an IP table file to assign tasks to each RPi. CDC weights are also created offline and loaded to the storage. In the case of a failure, the system uses another pre-defined distribution file with fewer devices that has a lower performance. In such a case, since the detection of a missing device takes time, the system mishandles many requests. Our proposes solution that has tolerance to such failures, so the system never loses a request. Additionally, with a close-to-zero recovery latency, the system proactively is more tolerant to straggler nodes.

**Weight Storage:** Each Pi has an SD card storage, for storing the weights, which is relatively inexpensive compared to the main memory. All trained weights are loaded to each Pi's storage (16 GB storage in our system), so each Pi can be assigned to execute any part of a layer. If local storage is limited, the assigned weight can also be shared on the network from a network-storage filesystem. This approach makes a tradeoff between how fast the switching time between different models can be and per-device storage usage. Additionally, note that the distribution method does not replace other methods, such as offloading to servers. The decision is the per-case basis and depends on several system-level decisions. The distribution offers the additional option of processing data locally.

### 6.1 System Recovery Case Studies

**Case Study I**: To depict the impact of how failures affect a system, we deploy AlexNet [40] on two IoT systems. The first system, shown in Figure 11a, contains five devices. The first fully-connected layer is split with the output-splitting method between two devices with no robustness method. The black bars in Figure 12 show the latency of the system when performing single-batch inferences. Now, if device C experiences failure, as shown in Figure 11b, other devices need to perform the task assigned to the failed device. Since the task of device C is the computation of half of the first fully-connected layer, device D needs to perform this extra task in addition to its task. After the failure is detected, which takes tens of seconds, the

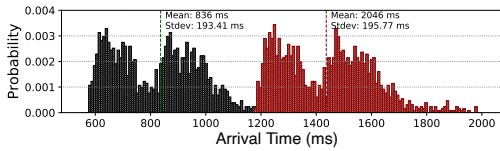

**Figure 12: Case study I: Recovery latency with & without CDC.**

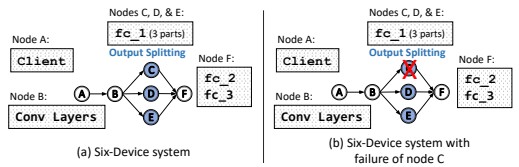

Figure 13: Case Study II: AlexNet on a six-device system.

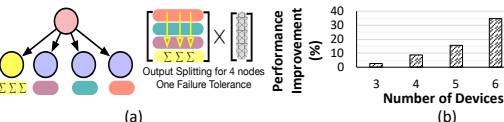

**Figure 16: Straggler mitigation study. (a) A system setup for four devices. (b) Straggler mitigation performance.**

red bars in Figure 12 depict the new shifted latency histogram of the system. Based on our measurements, on average, the system experiences 2.4x slowdown after recovery. The system is not performing beneficial work during failure detection, and experiences significant slowdown afterward. However, with our method, the system does not experience any slowdown or service interruption. **Case Study II**: As a remedy to failures, we deploy AlexNet on a six-device system. Figure 13a shows this system, in which an extra device is added for robustness using CDC. Note that our goal is to create robustness only for the first fully-connected layer and the extra device provides robustness to all the computations done on device D and E. If we experience failure, as Figure 13b shows, the performance of the system does not change. Additionally, during the operation without failure, we use the extra device to mitigate the straggler problem. Figures 14 and 15 show the system latency with and without this mitigation, respectively. As shown, the range and the distribution of latencies are improved towards a better performance. Thus, in addition to robustness, we can exploit the extra device to increase the performance.

## 6.2 Straggler Mitigation

We study straggler mitigation benefits by extending the previous system. To initiate recovery, a device waits for a particular amount of time. By adjusting this waiting threshold in a device, we can treat our method as a solution for the straggler problem after receiving the necessary amount of data. A lower threshold reduces latency and thus increases performance. Straggler problem is more prominent with more devices, so we set up an experiment as Figure 16a shows for a system with four devices, each of which performs a split in a fully-connected layer. Figure 16b shows performance improvement of straggler mitigation with a diffident number of devices in a system. The performance improvement is compared with the same system, with the same number of devices, with no straggler mitigation. As seen, for more devices, straggler mitigation has better performance (up to 35%) compared with a no-straggler-mitigation system with the same number of devices.

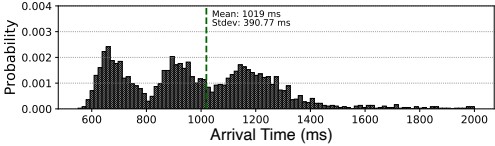

**Figure 14: AlexNet latency histogram without straggler mitigation.**

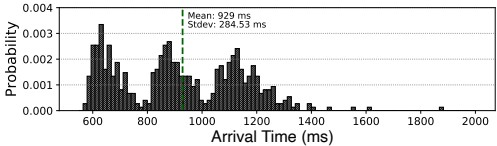

**Figure 15: AlexNet latency histogram with straggler mitigation.**

## 6.3 Full Model Coverage

In the system shown in Figure 13, devices with model parallelism are robust with CDC. For other devices, by the replication of the device's task (N-modular redundancy with $N = 2$, or 2MR), we can tolerate one failure in the entire system. Such a hybrid approach (CDC+2MR) could cover the entire system for failures. In a nutshell, our method covers any number of devices in one layer with just one additional device (this is for robustness to one failure). But, 2MR needs an additional device for each device. Therefore, our method has a *constant* cost with an additional number of devices; whereas 2MR requires a *linear* number of additional devices. In Figure 17, we study several DNNs [28, 45–47] with distributed implementations with tolerance to one failure with 2MR-only and CDC+2MR. Since CDC requires fewer devices than 2MR to cover the devices with model parallelism, the number of additional devices for full coverage for CDC+2MR is smaller than that of 2MR. The amount of difference depends on how many layers are distributed with model parallelism and how many devices are used per layer. For instance, Figure 17c and d depict two C3D distributions with different numbers of the devices for the layers that use model parallelism (two vs. three devices). We see that in Figures 17c and d, with two additional devices, CDC+2MR, compared with 2MR with 44% and 36%, reaches the coverage of 67% and 73%. This is because C3D distribution has two layers with model parallelism. Therefore, compared with 2MR, CDC+2MR achieves better coverage. In summary, if we use model-parallelism for a layer with $N$ number of devices, with $(1 + \frac{1}{N})$ times hardware cost, we can hide a single node's failure as opposed to 2x hardware cost in 2MR.

## 7 DISCUSSIONS

**The Introduced Computation:** The introduced new computations for our CDC-based method are similar to that of underlying GEMM computations. This is because we add new wights to the weight matrix (or a variant of it). These new weights can be calculated without the user's input and at a library level. Therefore, there are no additional costs for reprogramming the applications. Moreover, since the nature of the computations for these new weights is similar to that of DNNs, there is no need to design new kernels or distribution methods.

**Extending Robustness To More Failures:** Our discussions were focused on tolerating up to one failure. However, Extending to more than one failure is possible by adding new devices that perform computations based on the summation of some rows of weights instead of all of them. Figure 18 illustrates three setups in order of increasing tolerance to failures. The last setup tolerates two failures because new devices perform partial sums on the weights.[1] Thus,

---

[1]Note that the coverage to two failures is almost complete (partial error correction). We need Hamming-style coverage for full error correction.

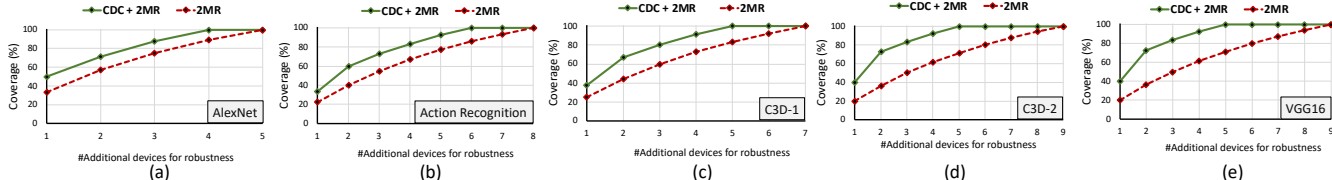

Figure 17: Full model coverage studies.

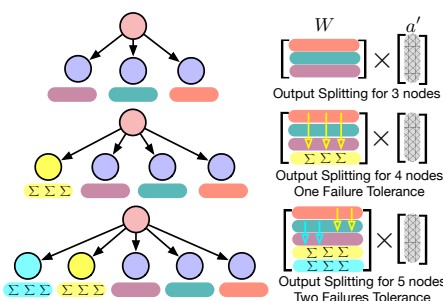

Figure 18: Tolerating multiple failures.

by utilizing idle devices with an overlapping set of weights, the robustness of the system increases.

## 8 RELATED WORK

With ubiquitous wireless networks and the availability of embedded processors, IoT devices are rapidly gaining ground [2–4]. The widespread everyday-life usage of IoT now includes smart thermostats, smart cameras, and personal assistants, and companies have introduced several IoT-based products and services [5, 6]. With the proliferation of IoT devices, a new computing paradigm using these devices has emerged as edge computing [7–9], in which the data processing is performed at the edge of the network. In the meantime, with the successful advancement of deep learning [1], many new and novel opportunities are created for IoT [7, 10–15, 28]. However, IoT devices have limited resources to execute heavy DNN models [7], and with such fast-paced advancements, the demanded computing power of DNN models is not expected to slow down.

To reduce the computing demand of DNNs, several techniques such as weight pruning [48], quantization and low-precision inference [49], and binarizing weights [50] are introduced to reduce the computations of DNNs. Several studies [10–12, 14, 15, 28, 51] also examined the distributed execution of DNNs on edge devices. Neurosurgeon [51] and Hauswald et al. [52] partition a DNN model between a single IoT and cloud. MoDNN [10] creates a local distributed mobile computing system and accelerates DNN computations on Galaxy S5. DDNN [11] also aims to enhance learning with multiple devices. In their work, the DNN model is retrained continuously while most of the computations are offloaded to the cloud. Hadidi et al. [12, 28] propose model parallelism methods, *but without discussing how to improve the robustness.*

The authors of coded distributed computing [23, 24] studies introduced coding for MapReduce-type workloads for large-scale computing. By coding, which increases the computation load of mapping functions, the amount of communication can be reduced in the reduction phase. The authors theoretically study the limits and tradeoffs of such distribution and illustrate an inverse relationship between the amount of computation and communication.

Usually, coding in CDC is applied over bit-level representation of numbers. *Instead of coding over floats/bits, our work applies coding to the application level by introducing new weights. Furthermore, in contrast, to reduce communication overhead in other studies, our goal is to increase robustness and tolerating unstable latencies.*

CDC helps to mitigate the straggler problem in computing clusters [53, 54], besides other methods such as straggler detection algorithms [55, 56] and replication-based approaches [57, 58]. Several works also utilize CDC to mitigate the straggler problem in distributed storage systems [59]. Distributed learning algorithms have also used CDC opportunities [25]. Since these algorithms use data parallelism for learning, CDC facilitates the mapping phase in learning algorithms with data shuffling. Particularly, Lee et al. [25] focused on two basic blocks of learning algorithms, matrix multiplication and data shuffling. *None of the above works has studied CDC in the context of robustness.* In contrast with our work, distributed learning studies [25] examine large-scale learning algorithms, which employ data parallelism, *whereas our work focuses on IoT-based inferencing, which utilizes model parallelism.*

## 9 CONCLUSION

In this paper, by utilizing CDC, we proposed a method to introduce tolerance for the single-batch inferencing of DNNs. Single-batch inferencing is important in IoT and near-the-edge computing domains because of the time-sensitivity of applications and the limited number of the requests in these domains. Our method exploits model-parallelism methods in prevalent DNN layers to add balanced computation for robustness. Model-Parallelism methods help us in achieving efficient system distribution by splitting the computation of single-batch inferencing among several IoT devices. We studied model-parallelism methods and their underlying computation when being distributed. To this end, we extended CDC to provide a trade-off between computations and robustness on distributed IoT-based systems.

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
