# OpenReview forum: "Creating Robust Deep Neural Networks With Coded Distributed Computing for IoT Systems"
_tinyml.org/tinyML/2021/Research_Symposium — Reject_

### Official Review · AnonReviewer3 · 2021-01-20

**Overall Merit Score:** 2

**Brief Summary:**

The paper provides different solutions to environments where devices (e.g. IOT devices) do not have adequate resource to provide for the compute (specifically CNN type workloads).  The paper provides multiple strategies on how to split the workload.  The paper provides specific examples for how to split of a matrix multiply workload across several devices.

**Detailed Comments:**

The paper presents a good solution to a specific problem of a set of nodes that does not have enough memory or compute to solve a larger problem.  The nodes have sufficient energy and sufficient communications.

The two assumptions of sufficient energy and sufficient communication may put it out of scope of most TinyML type problems.

**Paper Strengths:**

The paper provides very clear examples on how to split the workloads among devices.  The paper also show via simulation the effects of packet loss and latencies.

The author shows what the accuracy loss of a simple network (Lenet-5) and the loss of data on 1 layer.

**Paper Weaknesses:**

- The paper does not explore the impacts of power to communicate the activations (or values) across the different nodes vs. the power it would take to perform the compute.  The problem being solved is in an environment where there is limited compute and/or memory with more than sufficient power for the communications.  The example of a RPi used would generally not fit into this class of problems until the network got exceedingly large.
- The paper does not show how such a workload would be split among the end devices or how the devices could be provisioned.  .  Presumably, it could have been provisioned at installation.  Otherwise, the separate weights would have to be distributed post installation.  Provisioning at installation would also remove the problem of knowing nodes each of the devices would have to send its output to.
- The paper address pretty simple feed forward type of networks.  Networks with any branching (e.g. ResNet) would need additional compute and communications requirements and be more subject to degradation.

**Poster (If Paper Is Rejected):**

1: Yes, ok for poster sesion to nurture work

**Reviewer Confidence:**

4: The reviewer is confident but not absolutely certain that the evaluation is correct

---

### Official Review · AnonReviewer4 · 2021-01-29

**Overall Merit Score:** 2

**Brief Summary:**

The paper proposes a fault recovery method in a distributed IoT settings where additional nodes were introduced to produce summation of the input data as a means of redundancy. In case any node fails, the result could be recovered from additional nodes with only local subtraction in final device. The paper shows opportunities in model parallelisms (output, input, channel, filter and spatial) that could be exploited in distributed settings for redundancy/robustness.

**Detailed Comments:**

Most models (CNNs/DNNs) are heavily optimised for deployment on resource-constrained IoT devices. The addition of extra weights (through extra rows) will change the distribution of the layer-wise statistics on which many quantisation schemes depend. The paper should explore the scheme with real-life optimised and quantised int8 model. Without such a study, the real intricacies with low-precision distributed system are unknown.

The primary/basic scheme cannot handle more than one failure. To handle such scenarios, more nodes need to be added in the pool. In many applications, the user may not have any choice of adding an extra node. Also, the cost of adding nodes to cover many failure modes is very high.

The paper should compare their results with prior work.

In section 5.2, the authors state: "Hence, the distribution of these newly added computations follows the same rules and would not create an imbalance in the modified distribution."  But in section 5.3 they mentioned "...the robustness covers all devices, and the new computation does create an imbalanced distribution." These two statements seem to contradict. The author should clarify this statement.


**Paper Strengths:**

The paper is well written and covers many preliminaries in detail.
The full model coverage study is very detailed.


**Paper Weaknesses:**

The main assumption that all devices have access to the same input data reduces the applicability of this technique. In a distributed IoT scenario, it is most likely that each node will produce different data/input (based on location, direction, environment etc.). The research should focus on how to handle such scenarios when the full distribution is not available. For example, 2 out of 10 nodes are down and therefore ML task needs to deal with only 8 nodes. No additional node can produce such missing data.

The overhead from additional computation from additional rows is significant. For a distributed IoT system, adding 50% extra computation just for redundancy may not be acceptable. One could instead consider probabilistic techniques to deal with uncertainties from missing data.


**Poster (If Paper Is Rejected):**

1: Yes, ok for poster sesion to nurture work

**Reviewer Confidence:**

4: The reviewer is confident but not absolutely certain that the evaluation is correct

---

### Official Review · AnonReviewer1 · 2021-01-29

**Overall Merit Score:** 2

**Brief Summary:**

This paper considers the problem of distributing a matrix-matrix
multiplication computation of a DNN layer among IoT devices while being
able to tolerate these devices' unstable latencies and
intermittent failures. It is proposed to use a coded distributed
computing (CDC) scheme to provide redundancy in the matrix-matrix
multiplication, so IoT failures can be recovered in low latency.
To illustrate the method, experiments with distributed systems
comprising up to 12 Raspberry PIs are conducted.

**Detailed Comments:**

One would normally expect that decomposing a single matrix-matrix
multiplication over loosely connected, resource-constraining IoT
devices for distributed processing would incur a relatively high communication cost. This paper does not provide application scenarios and technical
results showing otherwise.

For the proposed CDC scheme, the paper would
benefit from further analysis and experimental results, demonstrating that communication cost could remain modest under
useful levels of redundancy provided by CDC.


**Paper Strengths:**

The proposed CDC mechanism is simple and could be of interest in
certain application scenarios.


**Paper Weaknesses:**

Distributing the computation of a single matrix-matrix
multiplication corresponding to a neural network layer to
multiple IoT nodes does not seem to be a reasonable
system approach. These methods will likely incur communication
overheads (such as latency and energy consumption), which could be
too large to be useful.  The paper does not address these issues;
there are no performance evaluations to estimate the communication
cost as a function of redundancy ensured by CDC.

**Poster (If Paper Is Rejected):**

1: No, paper is below bar for poster as well

**Reviewer Confidence:**

4: The reviewer is confident but not absolutely certain that the evaluation is correct

---

### Decision · Program_Chairs · 2021-02-05

**Decision:**

Reject

**Comment:**

Thank you for your submission.

Following careful consideration by our reviewers, we regret to inform you that we are unable to accept your submission.

Please refer to the reviewer comments for your reference. We hope you find this information helpful for submission to another venue, and we hope to see more of your work in the future.